# Spinal Irisin Gene Delivery Attenuates Burn Injury-Induced Muscle Atrophy by Promoting Axonal Myelination and Innervation of Neuromuscular Junctions

**DOI:** 10.3390/ijms232415899

**Published:** 2022-12-14

**Authors:** Sheng-Hua Wu, I-Cheng Lu, Shih-Ming Yang, Chia-Fang Hsieh, Chee-Yin Chai, Ming-Hong Tai, Shu-Hung Huang

**Affiliations:** 1Department of Anesthesiology, Kaohsiung Municipal Ta-Tung Hospital, Kaohsiung 801, Taiwan; 2Department of Anesthesiology, School of Medicine, College of Medicine, Kaohsiung Medical University, Kaohsiung 807, Taiwan; 3Department of Anesthesiology, Kaohsiung Medical University Hospital, Kaohsiung 807, Taiwan; 4Department of Anesthesiology, Kaohsiung Municipal Siaogang Hospital, Kaohsiung 812, Taiwan; 5Institute of Biomedical Sciences, National Sun Yat-Sun University, Kaohsiung 804, Taiwan; 6Department of Pathology, College of Medicine, Kaohsiung Medical University, Kaohsiung 807, Taiwan; 7Department of Pathology, Kaohsiung Medical University Hospital, Kaohsiung 807, Taiwan; 8Department of Surgery, Division of Plastic Surgery, Kaohsiung Medical University Hospital, Kaohsiung 807, Taiwan; 9Department of Surgery, School of Medicine, College of Medicine, Kaohsiung Medical University, Kaohsiung 807, Taiwan; 10Regeneration Medicine and Cell Therapy Research Center, Kaohsiung Medical University, Kaohsiung 807, Taiwan; 11Graduate Institute of Medicine, College of Medicine, Kaohsiung Medical University, Kaohsiung 807, Taiwan; 12Department of Surgery, Division of Plastic Surgery, Kaohsiung Municipal Siaogang Hospital, Kaohsiung 812, Taiwan

**Keywords:** irisin, burn injury, motor neuropathy, denervated muscle atrophy, neuromuscular junction

## Abstract

Muscle loss and weakness after a burn injury are typically the consequences of neuronal dysregulation and metabolic change. Hypermetabolism has been noted to cause muscle atrophy. However, the mechanism underlying the development of burn-induced motor neuropathy and its contribution to muscle atrophy warrant elucidation. Current therapeutic interventions for burn-induced motor neuropathy demonstrate moderate efficacy and have side effects, which limit their usage. We previously used a third-degree burn injury rodent model and found that irisin—an exercise-induced myokine—exerts a protective effect against burn injury-induced sensory and motor neuropathy by attenuating neuronal damage in the spinal cord. In the current study, spinal irisin gene delivery was noted to attenuate burn injury-induced sciatic nerve demyelination and reduction of neuromuscular junction innervation. Spinal overexpression of irisin leads to myelination rehabilitation and muscular innervation through the modulation of brain-derived neurotrophic factor and glial-cell-line-derived neurotrophic factor expression along the sciatic nerve to the muscle tissues and thereby modulates the Akt/mTOR pathway and metabolic derangement and prevents muscle atrophy.

## 1. Introduction

A burn injury is one of the most severe traumatic events and can result in debilitating morbidity and significant mortality and thus lead to a considerable economic and social burden. Burn injury leads to severe pain, increases infection and inflammation risk, and causes metabolic change; that may result in distributive shock, which leads to multiple organ failure [1]. However, >50% of patients with burn injury develop neuropathy that can last for several years, even after the burn wound heals completely [2,3]. Burn-related neuropathy mainly consists of sensory and motor neuropathy [4]. Patients with neuropathy often experience pain, paresthesia, or muscle weakness [4,5]. These are long-term sequelae that worsen the quality of life of the patients and lead to a higher economic burden for their families and medical burden for society [6]. The treatment of burn-related neuropathy remains challenging [4]. Current therapeutic interventions administered to patients with burn-related neuropathy involve gabapentinoids, antidepressants, steroids, and opioids for neuropathic pain and anabolic agents, including growth hormones, insulin-like growth factor I, insulin, corticosteroids, α/β-adrenergic blockers, and glucocorticoid antagonists, for burn-related muscle atrophy; however, the efficacy of these interventions is moderate, and they may have side effects, which limits their applicability [7,8,9,10,11]. Therefore, investigating the mechanisms underlying burn-related neuropathy development may facilitate the development of novel therapeutic approaches.

Burn injuries can cause systemic physiological dysregulation, which leads to hypermetabolism followed by muscle loss and weakness and then muscle atrophy [12]. Severe burn injury was also reported to activate apoptosis in skeletal muscle [13]. Furthermore, severe burn injury results in myocyte cell death and insufficient myogenesis [14], and denervation of the neuromuscular junction results in muscle atrophy [15]. Burn injury–induced motor neuropathy is associated with muscle wasting and weakness [4,5]. Early studies have reported that burn injury results in dysfunction of neuromuscular junctions and a denervation-like response in skeletal muscles [16,17,18,19]. A study demonstrated that severe burn injury promotes spinal microglia activation, resulting in motor neuron degeneration and, consequently, decreased neuromuscular junction innervation and subsequent muscle atrophy [20]. Although burn injury results in metabolic homeostasis disruption followed by muscle atrophy, the molecular mechanism underlying burn injury-induced motor neuropathy and subsequent neuromuscular junction denervation–induced atrophy has yet to be elucidated.

Irisin was primarily reported to be an exercise-induced hormone that is mainly secreted after cleavage from fibronectin type III domain-containing protein 5 (FNDC5) by skeletal muscle cells [21,22]. Irisin functions as a fat-browning myokine, which alleviates metabolic syndromes, such as insulin resistance, type-2 diabetes mellitus, and obesity [21,23]. Irisin also has been reported to have multiple functions in various cell types [24]. The most notable of these functions are its neuroprotective and anti-neuroinflammatory effects [25,26]. Irisin was reported to be an essential factor for neural differentiation in mouse embryonic stem cells [27]. In addition, patients with Alzheimer’s disease and mice with neurodegenerative disease were reported to have reduced FNDC5 and irisin levels in their cerebrospinal fluid (CSF) and hippocampus, and in these patients and mice, FNDC5 and irisin expression restoration was noted to improve synaptic plasticity and memory [28]. Irisin protects neurons from ischemia/reperfusion injury-induced apoptosis, due to its suppressive effects on microglia activation and monocyte infiltration [29,30]. Exercise and physical therapy have been reported to alleviate neuropathic pain and burn-related muscle atrophy [31,32,33,34,35]. As a myokine, irisin has also been suggested to alleviate pain sensitization [36,37].

In our previous study, we observed that irisin gene delivery ameliorated burn-induced sensory and motor neuropathy by reducing tumor necrosis factor α–induced neuronal apoptosis [38], which indicates that irisin has antineurodegenerative effects on burn-induced neuropathy. Specifically, we reported that intrathecal (i.t.) adenovirus irisin (Ad-irisin) ameliorated burn-related neuropathy by restoring decreased spinal FNDC5/irisin expression and reduced CSF irisin levels, which attenuated burn-induced spinal injury and subsequent pain sensitization and muscle atrophy [38]. However, the effects of spinal irisin gene delivery on motor neuronal regeneration and neuromuscular junction recovery promotion after burn injury remain unknown. In the present study, we evaluated whether spinal irisin gene delivery promotes myelination and neuromuscular junction recovery and subsequently prevents muscle loss and weakness.

## 2. Results

### 2.1. Spinal Irisin Gene Delivery Attenuated Burn-Induced Muscle Weakness and Loss

We used our rodent burn model to validate whether spinal irisin gene delivery improves burn-related muscle atrophy (Figure 1A). Our results indicated that i.t. Ad-irisin attenuated burn-induced muscle weakness as well as provided moderate rehabilitation of grip strength (Figure 1B).

Muscle atrophy typically occurs as a consequence of abnormal metabolism in which muscle protein is replaced with fat and glycogen [39]. Glycogen overaccumulation can reduce muscle endurance [40]. Through glycogen detection as well as PAS and Oil Red O staining, we observed that i.t. Ad-irisin moderately ameliorated burn-induced accumulation of lipid and glycogen in ipsilateral gastrocnemius muscle (Figure 1C,E). Next, we analyzed the expression of atrophy-associated proteins and observed that i.t. Ad-irisin attenuated burn-induced increases in muscular atrogin-1 and muscle ring-finger protein-1 (MuRF-1; Figure 2A–D,F,G), both of which are well-documented muscle atrophy markers [41]. Moreover, i.t. Ad-irisin attenuated the post-burn decreased expression of muscular irisin (Figure 2D,E).

As a promyogenic factor, recombinant irisin rescues denervation-induced atrophy when intraperitoneally injected after notexin-induced neuronal damage [42]. This result suggests that spinal irisin gene delivery promotes rehabilitation of neuromuscular junctions after burn injury. Therefore, we analyzed survival motor neuron (SMN) expression in ipsilateral gastrocnemius muscle after burn injury. We observed that burn injury resulted in decreased SMN expression in the ipsilateral gastrocnemius muscle and that spinal irisin gene delivery attenuated the burn-induced decrease in SMN expression (Figure 2D,H). This result indicates that i.t. Ad-irisin promoted rehabilitation of burn-induced neuromuscular junction denervation.

### 2.2. Spinal Irisin Gene Delivery Attenuated Burn-Induced Inactivation of Akt/mTOR Signaling and Enhancement of Autophagic Markers in Skeletal Muscle

Burn-induced muscle wasting typically accompanies metabolic dysregulation. The Akt/mTOR pathway—a key regulatory signaling pathway involved in the modulation of both catabolism and anabolism—is a crucial pathway regulating muscle hypertrophy and atrophy [43]. Muscle atrophy has been noted to downregulate this pathway [43]. We therefore analyzed the expression of Akt/mTOR pathway proteins in ipsilateral gastrocnemius muscle after burn injury. The results demonstrated that burn injury induced Akt/mTOR pathway inactivation and reduced the p-Akt/Akt and p-mTOR/mTOR ratios and that i.t. Ad-irisin recovered burn-induced inactivation of the Akt/mTOR pathway (Figure 3A–E). Moreover, overall mTOR expression decreased after burn injury, whereas i.t. Ad-irisin attenuated burn-induced mTOR downregulation (Figure 3A,D,E).

The autophagy–lysosome pathway is activated during catabolic conditions in muscle cells [44]. Therefore, we determined the expression of autophagy markers. The results indicate that burn injury resulted in increased beclin 1, LC3A, and LC3B expression and that i.t. Ad-irisin attenuated these increases (Figure 3A,D–F).

### 2.3. Spinal Irisin Gene Delivery Rehabilitated Myelination and Neuromuscular Junctions following Burn Injury

We previously reported that spinal irisin gene delivery ameliorates burn injury-induced neuronal damage in the ventral horn [38]. In the current study, i.t. Ad-irisin was noted to promote rehabilitation of denervation in neuromuscular junctions, indicating that the ameliorative effects of irisin gene delivery involve protective effects on the neuronal axis along the spinal cord to peripheral tissues following burn injury. In patients with burns, burn injury results in demyelination of the neuronal axon [5,45]. Therefore, we determined whether irisin gene delivery protects neurons from burn-induced demyelination by observing the myelin sheath of the neuronal filaments of the sciatic nerve. The results demonstrate that i.t. Ad-irisin attenuated burn injury-induced Schwann cell immaturation, increased p75NTR, and reduced MBP levels (Figure 4A–C).

We subsequently investigated whether spinal irisin gene delivery attenuates burn-induced denervation of the neuromuscular junction of ipsilateral gastrocnemius muscle by using a fluorophore-conjugated α-bungarotoxin (α-BTX), a toxic snake venom alpha-neurotoxin that binds to acetylcholine receptor (AChR) at the synapse [46], and by costaining for the neurofilament markers synapsin and NF200. We determined the distribution of the motor neurons and neuromuscular junctions on the ipsilateral gastrocnemius muscle and found that burn injury reduced synapse activity and led to weak α-BTX staining; in addition, i.t. Ad-irisin attenuated burn injury, resulting in decreased neuromuscular junction function (Figure 4D,E).

### 2.4. Spinal Irisin Gene Delivery Modulated Expressions of BDNF and GDNF in Ventral Horn and Sciatic Nerve

Irisin can regulate neurotrophic factors, including brain-derived neurotrophic factor (BDNF) and glial-cell-line-derived neurotrophic factor (GDNF) [47,48,49,50]. BDNF promotes motor function recovery and myelination during nerve neogenesis [51,52]. Compared with BDNF, GDNF demonstrates a higher potential for the improvement of functional reinnervation of motor axons in injured nerves [53]. To evaluate the regulatory effects of irisin, we evaluated BDNF and GDNF expression in the spinal cord and sciatic nerve. We discovered that burn injury resulted in BDNF upregulation and i.t. Ad-irisin attenuated burn-induced BDNF overexpression in the ventral horn and sciatic nerve (Figure 5A,B,D,E). Moreover, we found that burn injury did not result in a GDNF expression change in the ventral horn with or without i.t. Ad-irisin (Figure 5A,C). However, burn injury resulted in decreased GDNF expression in the sciatic nerve, and i.t. Ad-irisin attenuated the decreased expression (Figure 5D,F). These results indicate that spinal irisin overexpression regulates BDNF and GDNF expression between the spinal cord and peripheral nervous system in a distinct manner.

### 2.5. Spinal Irisin Gene Delivery Modulated Expressions of BDNF and GDNF in Skeletal Muscle

BDNF and GDNF, which have mainly been reported as neuronal factors, also function in non-neuronal tissues, particularly skeletal muscles. BDNF elicits functional inhibition of myogenic differentiation in satellite cells or muscle progenitor cells [54]. In addition, GDNF improves motor neuron survival and function [55], and elevated GDNF expression in skeletal muscles promotes neuromuscular junction innervation [56]. We therefore analyzed BDNF and GDNF expression in ipsilateral gastrocnemius muscle after burn injury with or without i.t. Ad-irisin through immunofluorescence staining and immunoblotting analyses. The results indicate that burn injury led to BDNF overexpression in ipsilateral gastrocnemius muscle (Figure 6A,B) and that i.t. Ad-irisin attenuated this overexpression. Specifically, burn injury resulted in proGNDF and mGDNF downregulation in ipsilateral gastrocnemius muscle (Figure 6A,D,G,H), and i.t. Ad-irisin attenuated burn-induced proBDNF and mBDNF expression (Figure 6D–F).

## 3. Discussion

Burn injury-related morbidity remains a serious clinical issue. Burn survivors often develop sensory and motor neuropathy with prolonged pain sensitization and physical disability, even after the burn wound heals. Moreover, burn injury can induce muscle loss and weakness. Burn-induced atrophy is a consequence of the dysregulation of metabolic homeostasis and inflammatory responses, cachexia and sarcopenia with increased protein degradation and muscle cell death, and a reduction in protein synthesis and muscle cell growth [57]. In addition, burn injury results in motor neuropathy represented by motor neuronal damage or neuromuscular dysfunction associated with muscle atrophy [20]. However, the mechanisms underlying neuromuscular junction dysfunction and neuronal axon degeneration after burn injury remain unclear.

Although the contribution of muscle denervation and axonal degeneration to muscle loss and weakness has been well-characterized [58,59], the mechanism underlying burn injury-induced dysfunction or dysregulation of motor neurons warrants elucidation. We previously reported that burn injury promotes apoptosis of both neuronal cells in the ventral horn and Schwann cells in the sciatic nerve [38,60]. In addition, Ma et al. reported that motor neuron degeneration due to burn injury promotes increased proinflammatory cytokine expression, which thereby promotes microglia activation [20,38]. In the present study, we validated that burn injury results in sciatic nerve demyelination (Figure 4A–C) and that Schwann cell dedifferentiation increases the expression of the immature marker p75 and reduces that of the maturated marker MBP (Figure 4A–C) [61,62]. Spinal gene delivery attenuated burn-induced Schwann cell degeneration as well as both p75 and MBP expression, indicating that spinal irisin gene delivery recovers burn-induced demyelination (Figure 4A–C). Furthermore, by using staining with α-BTX, we found that i.t. Ad-irisin attenuated burn-induced decreases in muscle innervation and reduced the number of neuromuscular junctions (Figure 4D,E). Our results demonstrate that third-degree burn injury not only damages neuronal bodies in the ventral horn but also causes axonal damage, as represented by demyelination and a reduction in neuromuscular junctions as well as synapses in ipsilateral gastrocnemius muscle. Spinal irisin gene delivery attenuated burn-induced damage to the neurons in the ventral horn by reducing axonal myelination and neuromuscular junctions (Figure 4D,E).

Burn injury results in prolonged dysregulation of metabolic homeostasis and inflammatory responses, which results in muscle atrophy [57]. Elevation in overall energy expenditure leads skeletal muscle protein breakdown rates to surpass synthesis rates [63]. Burn injury also results in ectopic fat deposition in the vital organs, particularly the liver and skeletal muscles [64]. Kaur et al. reported that burn injury promotes metabolic derangements with overexpression of adipose triglyceride lipase, which is responsible for the breakdown of fat stores into free fatty acids and glycerol in adipose tissue. The breakdown of fat stores elevates circulating fatty acid levels and shuttles fat to the liver, resulting in fatty liver development [65]. Through proton nuclear magnetic resonance spectroscopy, Astrakas et al. determined that in addition to fatty liver, burn injury leads to localized intramyocellular lipid accumulation in skeletal muscles [66]. Our current findings also demonstrate that burn injury results in lipid accumulation and increased Oil Red O staining intensity in ipsilateral gastrocnemius muscle. We also observed that i.t. Ad-irisin attenuates lipid accumulation (Figure 1D,F). Abdullahi et al. reported that blockade of interleukin 6 signaling attenuated burn-induced fat browning and hepatic steatosis [67]. These results suggested that increased overall inflammation is associated with ectopic fat deposition. Li et al. indicated that chronic low-grade inflammation reduces the lipid storage capacity of adipose tissue by inhibiting preadipocyte differentiation and increasing lipolysis, which consequently increases ectopic fat deposition [68,69]. However, the contribution of inflammation to muscle fat deposition remains unclear [68,69]; this may be because evidence regarding metabolic dysregulation and inflammatory responses after burn injury is insufficient. Nevertheless, in the contexts of chronic damage and inflammation in skeletal muscle, muscle fat deposition was determined to lead to the transdifferentiation of muscle stellate cells to adipocytes [70]. Yousuf et al. reported that metformin ameliorates burn-induced atrophy by enhancing a myogenic phenotype by affecting Pax7-positive skeletal muscle progenitor cells [71]. Moreover, synergistic crosstalk between neuronal and muscular progenitor cells in the mesenchymal niche is essential for functional neuromuscular regeneration and maturation in both neuromuscular junction and skeletal muscle cells [72]. Our results demonstrate that spinal irisin gene delivery attenuates burn injury, which results from neuromuscular junction dysregulation caused by the interplay between the co-maturation of muscle and motor neuron progenitor cells in the muscle regenerative process after burn injury.

We also observed that burn injury results in decreased Akt/mTOR pathway activity and that spinal irisin gene delivery promotes Akt/mTOR pathway activation in ipsilateral gastrocnemius muscle (Figure 3). Akt/mTOR pathway activation is a crucial regulator that prevents skeletal muscle atrophy [43]. However, tightly controlled activity of mTOR has been reported its importance in homeostasis of skeletal muscle toward either anabolism or catabolism, resulting in hypertrophy or atrophy modulation [73]. In contrast to burn-induced muscle atrophy, mTOR activation promotes atrophy when motor neuron damage is severe [74,75]. Castets et al. reported that in a sciatic nerve transection model, Akt/mTOR pathway activation was required to maintain muscle homeostasis in denervated skeletal muscles [74]. Tang et al. reported that in a sciatic nerve transection model, mTORC1 activation contributed to muscle atrophy exacerbation [75]. Further investigation elucidating the role of mTOR signaling in muscle homeostasis in severe motor neuron injury and burn-induced motor neuropathy is warranted. Nakazawa et al. reported that burn injury results in metabolic derangements and mitochondrial dysfunction of skeletal muscle through the activation of both mTORC1 and HIF-1α after 3 days of burn injury on 30% of the total body surface area [76]. By contrast, we observed decreased Akt/mTOR signaling activity in the fourth week after burn injury even when the burn wound was healed (Figure 3). This result indicates that Akt/mTOR pathway modulation after burn injury might involve dynamic regulation of muscular homeostasis during the process of healing and continuous care after burn injury. Further investigation to elucidate Akt/mTOR pathway modulation in the acute and rehabilitative phases is warranted.

The regulatory effects of irisin on BDNF and GDNF expression have been reported in many studies [47,48,49,50]. Both BDNF and GDNF have crucial roles in neuronal regeneration in terms of both neuronal survival and axonal growth [77]. However, the regenerative effects of BDNF and GDNF lead to different patterns of recovery in injured motor neurons [77]. After a nerve crush and transection, BDNF and full-length TrkB receptor expression increase in the motor neurons [78]; however, this increase occurs transiently, until up to 7 days after injury [79,80]. Moreover, although BNDF promotes transient recovery of neuronal injury, prolonged exogenous treatment has disadvantages. Novikov et al. reported that treatment with recombinant BDNF promoted axonal regeneration of motor neurons in a rodent ventral root evulsion model. However, long-term (12-week) treatment with recombinant BDNF resulted in a loss of S-type boutons [81]. In addition, intraspinal axonal sprouting induced by BDNF treatment can result in spasticity, which may restrain the therapeutic effects of BDNF [82]. In contrast to severe damage and nerve transection, burn injury-induced motor neuron damage may mainly involve the modulation of myelination [5,45]. However, by using a full-length TrkB receptor, Xiao et al. discovered that BDNF inhibits myelination [83].

Eggers et al. reported that GDNF promotes reinnervation of motor axons but negatively regulates Schwann cells from maturation and myelination [84,85]. In contrast to the tissue-specific delivery of GDNF in nerve tissue, systemic GDNF treatment promotes myelination of fibers that are normally nonmyelinated [86]. Our results demonstrate that spinal irisin gene delivery does not result in significant changes in GDNF expression in the ventral horn, whereas after rehabilitation, burn injury induces a decrease in GDNF expression in the sciatic nerve (Figure 5D,F). Our previous results demonstrated that spinal irisin gene delivery prevents apoptosis of the ventral horn neurons after burn injury [38]. These results indicate that spinal irisin gene delivery-promoted myelination might not occur through the modulation of GDNF. Jiang et al. reported that intravenous injection of recombinant irisin reversed spinal cord injury-induced demyelination [87]. However, further comprehensive investigation of the dominant effects of irisin and neurotrophins on neuronal regeneration and myelination is warranted. Our present results reveal that after 1 week, spinal irisin gene delivery can regulate BDNF and GDNF expression in both the ventral horn and sciatic nerve and can modulate p75NTR and MBP expression in the sciatic nerve (Figure 4A–C and Figure 5). This result indicates that increased spinal irisin expression modulates BDNF and GDNF expression, which mainly involves Schwann cell maturation and myelination, after burn-induced injury. To understand the therapeutic potential of spinal irisin treatment, further investigation of the long-term, mechanistic effects of irisin on BDNF and GDNF regulation and the effects of irisin alone after burn-induced motor neuropathy is warranted. BDNF and GDNF expression modulate neuromuscular junctions [88,89]. Song et al. reported that BDNF suppressed neuromuscular junction formation and maturation [88]. Nguyen et al. reported that GDNF overexpression in muscles resulted in neuromuscular junction hyperinnervation [89]. Our results demonstrate that burn injury results in increased expression of both pro-BDNF and mBDNF and reduced expression of both pro-GDNF and mGNDF. In addition, i.t. Ad-irisin attenuated these changes in expression (Figure 5 and Figure 6). These results indicate that spinal irisin gene delivery promotes rehabilitation of burn-induced reductions in neuromuscular junctions, and the underlying mechanism might involve BNDF and GDNF regulation in skeletal muscles. Furthermore, pro-BDNF was reported binding to p75^NTR^ and promotes retraction of neuromuscular junctions, whereas mBDNF was reported binding TrkB and promotes both extension of motor neurons and stabilizes neuromuscular junctions [90]. The balance between pro-BDNF and mBDNF levels interplays formation of functional neuromuscular junctions [90]. We found that spinal irisin gene delivery attenuated burn injury-induced increase of pro-BDNF in ipsilateral gastrocnemius muscle, but no significant change of mBDNF level was found (Figure 6). These results indicate that burn injury-induced expressional change of both pro-BDNF and mBDNF might be involved in formation of functional neuromuscular junctions. Spinal irisin gene delivery normalized balance between pro-BDNF and mBDNF might be involved in rehabilitation of neuromuscular junctions after burn injury.

## 4. Materials and Methods

### 4.1. Animals, Full-Thickness Burn Injury Model, and Grip Strength Measurement

Adult male Sprague Dawley rats were obtained from BioLASCO Taiwan (Taipei, Taiwan) and maintained in specific pathogen-free animal facilities with water and commercial rat food provided ad libitum under a 12-h light–dark cycle. Our experimental design was approved by the Institutional Animal Care and Use Committee of Kaohsiung Medical University (approval numbers: 108195, 108132, and 109127).

A full-thickness burn injury model was induced in these rats as described previously [38]. In brief, we introduced a third-degree burn wound on the right hind paws of the rats and then applied silver sulfadiazine cream for 3 weeks to allow for the burn wound to heal. The average related strength level was determined as the ratio of the ipsilateral to the contralateral hindlimb strength levels measured three times on a grip strength meter (BIO-GS3; Bioseb, Pinellas Park, FL, USA).

### 4.2. Adenovirus and Intrathecal Injection

Recombinant adenovirus containing the irisin sequence with a signal peptide on the N’ terminal (Ad-irisin) was generated as described previously [38]. Intrathecal injection of recombinant adenovirus was carried out at post-burn week 3 as described previously [38]. Phosphate-buffered saline and Ad-GFP injections were used as injection and viral infection controls, respectively.

### 4.3. Immunofluorescence Staining, α-Bungarotoxin Staining, and Oil Red O Staining

Frozen tissues were sliced into 5 μm thick sections and permeabilized using a buffer containing 1.5% normal serum with 0.2% triton X-100 in TBST. After incubation with antibodies, Alexa Fluor 555-conjugated α-bungarotoxin (Thermo Fisher Scientific, Waltham, MA, USA), and nuclear staining dye, the sections were mounted with an antifade medium and visualized under a fluorescence microscope.

The sections were also stained using Oil Red O staining after fixation using formalin. In brief, after they were rinsed with 60% isopropanol, the slices were incubated with Oil Red O working solution and hematoxylin. After a wash with distilled water, the slices were mounted and visualized under a light microscope.

The immunofluorescence and Oil Red O staining intensities were quantified using Image J (NIH, Bethesda, MD, USA).

### 4.4. Periodic Acid–Schiff Staining

We subjected 3 μm thick paraffin-embedded tissue sections to deparaffinization and rehydration. They were then incubated with 0.5% periodic acid and washed with distilled water. Next, the sections were incubated with Schiff reagent and washed with lukewarm water. After they were counterstained with hematoxylin, the sections were dehydrated and visualized under a light microscope. The staining intensity was quantified using Image J (NIH).

### 4.5. Immunoblotting Analysis

After homogenization using RIPA buffer containing a protease and phosphatase inhibitor (Roche, Basel, Switzerland), the protein extract was separated through electrophoresis through sodium dodecyl sulfate polyacrylamide gel electrophoresis (Bio-Rad, Burlington, MA, USA) and transferred onto polyvinylidene fluoride transfer membranes. After they were blocked with 5% skim milk, the membranes were incubated with a primary antibody and horseradish peroxidase-conjugated secondary antibody. The signals were visualized and exposed using X-ray film (Fujifilm Corporation, Kyoto, Japan), and the signal density and intensity were determined using ImageJ (NIH).

### 4.6. Glycogen Detection

Gastrocnemius muscle tissue was weighed and homogenized following alkalizing and acidifying reactions. After participating, crude glycogen was extracted in 95% ethanol. The glycogen was detected followed the instructions of the manufacturer (Abcam, Boston, MA, USA).

### 4.7. Antibodies

Commercial antibodies used for immunofluorescence staining and immunoblotting analysis are listed following: Atrogin-1 (DF7075, Affinity Biosciences, Jiangsu, China), MuRF (bs-2539R, Bioss Antibodies, Woburn, MA, USA), P75^NTR^ (ABN1655, Merck Millipore, Darmstadt, Germany), MBP (AB5864, Merck Millipore, Darmstadt, Germany), Synapsin (60191-1-Ig, Proteintech, IL, USA), NF200 (N5389, Merck Millipore, Darmstadt, Germany), BDNF (bs-4989R, Bioss Antibodies, Woburn, MA, USA), GDNF (JM-5098-3, MBL INTERNATIONAL, Woburn, MA, USA), SMN (NB100-1936, Novus Biologicals, CO, USA), GADPH (G8795, Sigma-Aldrich, St. Louis, MO, USA), pAKT (4060, Cell Signaling Technology Inc., Beverly, MA, USA), AKT (9272, Cell Signaling Technology Inc., Beverly, MA, USA), p-mTOR (5536, Cell Signaling Technology Inc., Beverly, MA, USA), mTOR (2983, Cell Signaling Technology Inc., Beverly, MA, USA), beclin1 (3738, Cell Signaling Technology Inc., Beverly, MA, USA), and LC3B (2775, Cell Signaling Technology Inc., Beverly, MA, USA).

### 4.8. Statistical Analysis

Student’s *t*-test was used to identify between-group differences. All values are expressed as the mean ± standard deviation, and a *p* value of <0.05 was considered to indicate statistical significance.

## 5. Conclusions

In this study, we found that spinal irisin gene delivery attenuates burn injury-induced neuromuscular junction demyelination and degeneration. Increased spinal irisin levels promote the rehabilitation of motor neuronal function and recovery of functional neuromuscular junctions, which results in the alleviation of burn-induced muscle weakness and loss. The post-burn injury effects of spinal irisin gene delivery might involve motor neuron death prevention, sciatic nerve demyelination, and decreased neuromuscular junction innervation. The rehabilitation of functional neuromuscular junctions may alleviate metabolic derangement of skeletal muscle, including lipid deposition and increased glycogen accumulation, and the consequent muscle weakness and loss. The regulatory effects of spinal irisin gene delivery in the rehabilitation of motor neuronal functions after burn injury involve modulation of BDNF and GDNF expression and attenuation of burn injury-induced motor neuropathy and consequent muscle loss and weakness.

## Figures and Tables

**Figure 1 ijms-23-15899-f001:**
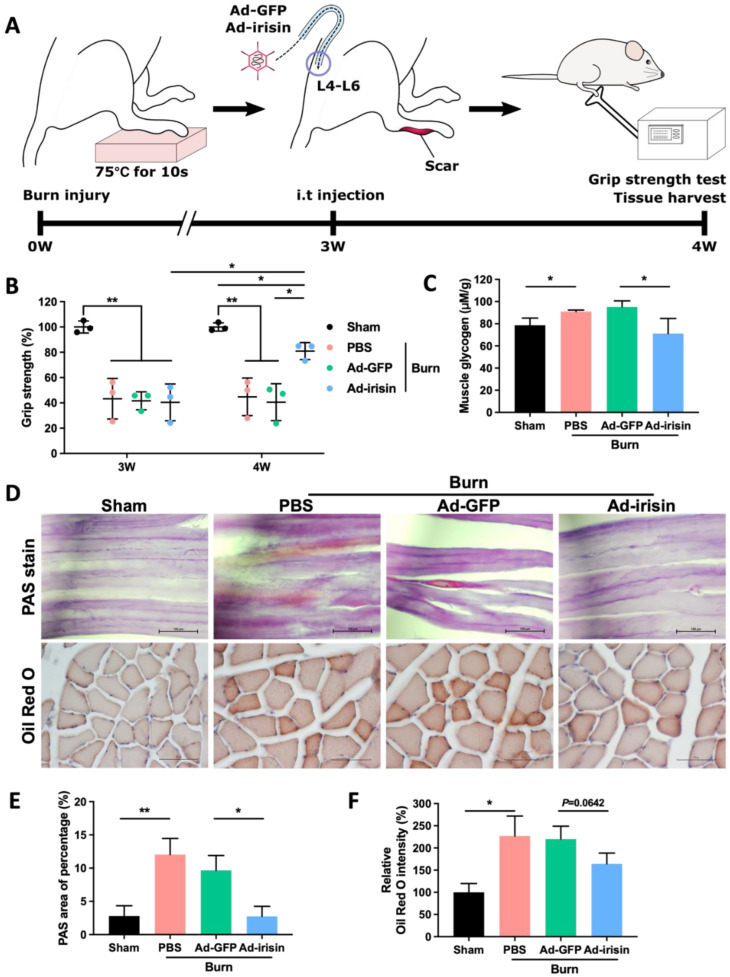
Spinal irisin gene delivery rehabilitated burn injury-induced muscle weakness and attenuated metabolic derangement. (**A**) Full-thickness burn injury was induced on the right hind paws of rats on a heated metal block (75 °C ± 0.5 °C) for 10 s. Silver sulfadiazine cream was applied to their paws every day for 3 weeks until their wounds closed completely. Intrathecal (i.t.) administration of Ad-GFP and Ad-irisin (1 × 108 c.f.u.) was performed in the third week after burn induction. Grip strength was measured at the third week, right before the i.t. injection, and in the fourth week, as a post-injection, for 1 week. Tissues were harvested for further analyses. (**B**) Representative scatterplot illustrating the ratio of the related ipsilateral to contralateral hindlimb grip strength after burn induction in the third and fourth weeks. These ratios were compared with those of a sham group at each time point. Error bars represent standard deviations (SDs). * *p* < 0.05, ** *p* < 0.01, unpaired *t*-test. (**C**) Representative bar graph of glycogen levels in ipsilateral gastrocnemius muscle detected using a glycogen detection kit (Abcam, Boston, MA, USA) in the fourth week after burn injury. Error bars represent SDs. * *p* < 0.05, ** *p* < 0.01, unpaired *t* test. (**D**) PAS and Oil Red O staining of ipsilateral gastrocnemius muscle in the fourth week after burn injury. (**E**) Representative bar graph illustrating percentage of PAS-positive area. Error bars represent SDs. * *p* < 0.05, ** *p* < 0.01, unpaired *t* test. (**F**) Representative bar graph illustrating relative intensity of Oil Red O staining. Error bars represent SDs. * *p* < 0.05, unpaired *t*-test.

**Figure 2 ijms-23-15899-f002:**
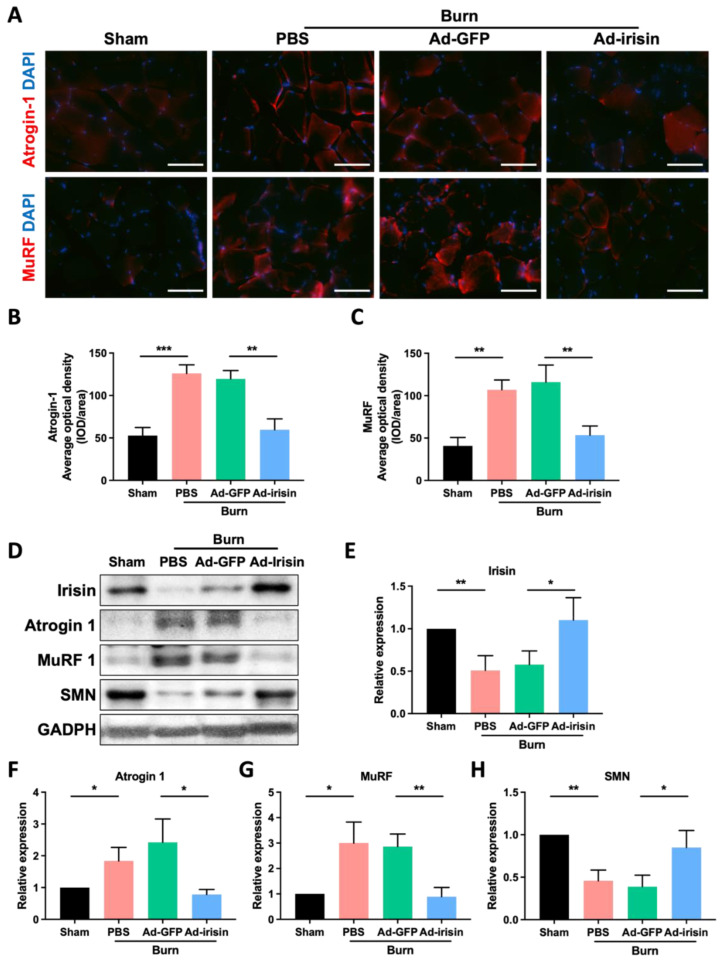
Spinal irisin gene delivery attenuated burn injury-induced atrophy markers and reduced irisin and SMN expression in ipsilateral gastrocnemius muscle. (**A**) Immunofluorescence staining of atrogin-1 and MuRF in ipsilateral gastrocnemius muscle in the fourth week after burn injury. DAPI counterstain was used to locate the nucleus. (**B**,**C**) Representative bar graphs illustrating averaged optical intensity of atrogin-1 and MuRF. Error bars represent standard deviations (SDs). ** *p* < 0.01, *** *p* < 0.001, unpaired *t*-test. (**D**) Immunoblot of ipsilateral gastrocnemius muscle tissue in the fourth week after burn injury. (**E**–**H**) Representative bar graphs illustrating normalized expression ratio of irisin, atrogin-1, MuRF, and SMN, with GAPDH as an internal control. Error bars represent SDs. * *p* < 0.05, ** *p* < 0.01, unpaired *t*-test.

**Figure 3 ijms-23-15899-f003:**
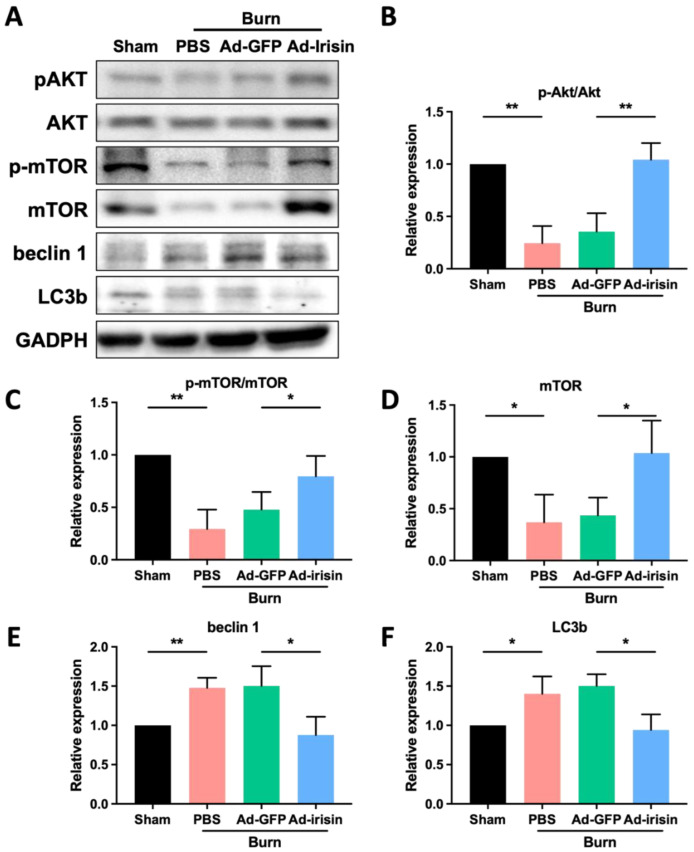
Spinal irisin gene delivery attenuated decreased Akt/mTOR pathway activity. (**A**) Tissue of ipsilateral gastrocnemius muscle in the fourth week after burn injury was subjected to immunoblotting analysis. (**B**) Representative bar graph illustrating the normalized p-Akt/Akt expression. Error bars represent standard deviations (SDs). ** *p* < 0.01, unpaired *t*-test. (**C**) Representative bar graph illustrating the normalized p-mTOR/mTOR expression. Error bars represent SDs. * *p* < 0.05, ** *p* < 0.01, unpaired *t*-test. (**D**–**F**) Representative bar graphs illustrating the normalized expression of mTOR, beclin 1, and LC3b, with GAPDH as an internal control. Error bars represent SDs. * *p* < 0.05, ** *p* < 0.01, unpaired *t*-test.

**Figure 4 ijms-23-15899-f004:**
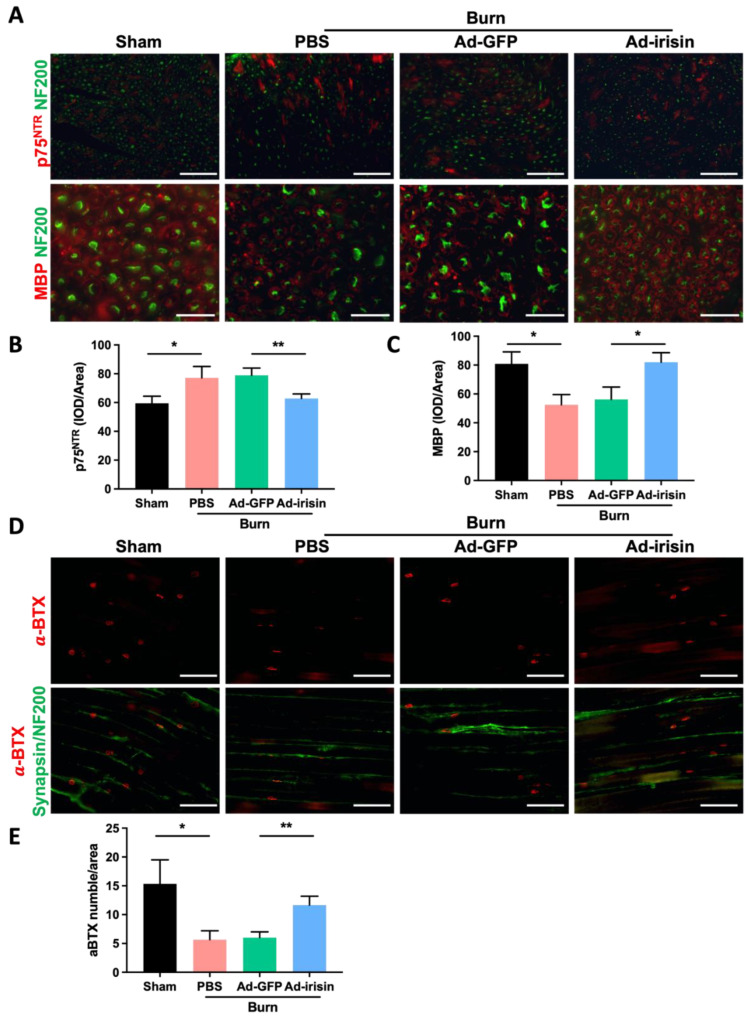
Spinal irisin gene delivery attenuated burn injury-induced demyelination in the sciatic nerve and decreased innervation of neuromuscular junction. (**A**) Immunofluorescence staining of p75NTR and MBP in the sciatic nerve in the fourth week after burn injury. NF200 stain was used to visualize neurofilaments. (**B**,**C**) Representative bar graphs illustrating averaged optical intensity of p75NTR and MBP. Error bars represent standard deviations (SDs). * *p* < 0.05, ** *p* < 0.01, unpaired *t*-test. (**D**) Immunofluorescence staining of α-BTX in ipsilateral gastrocnemius muscle in the fourth week after burn injury. Synapsin and NF200 staining was used to visualize neurofilaments. (**E**) Representative bar graph illustrating the number of α-BTX-positive areas. Error bars represent SDs. * *p* < 0.05, ** *p* < 0.01, unpaired *t* test.

**Figure 5 ijms-23-15899-f005:**
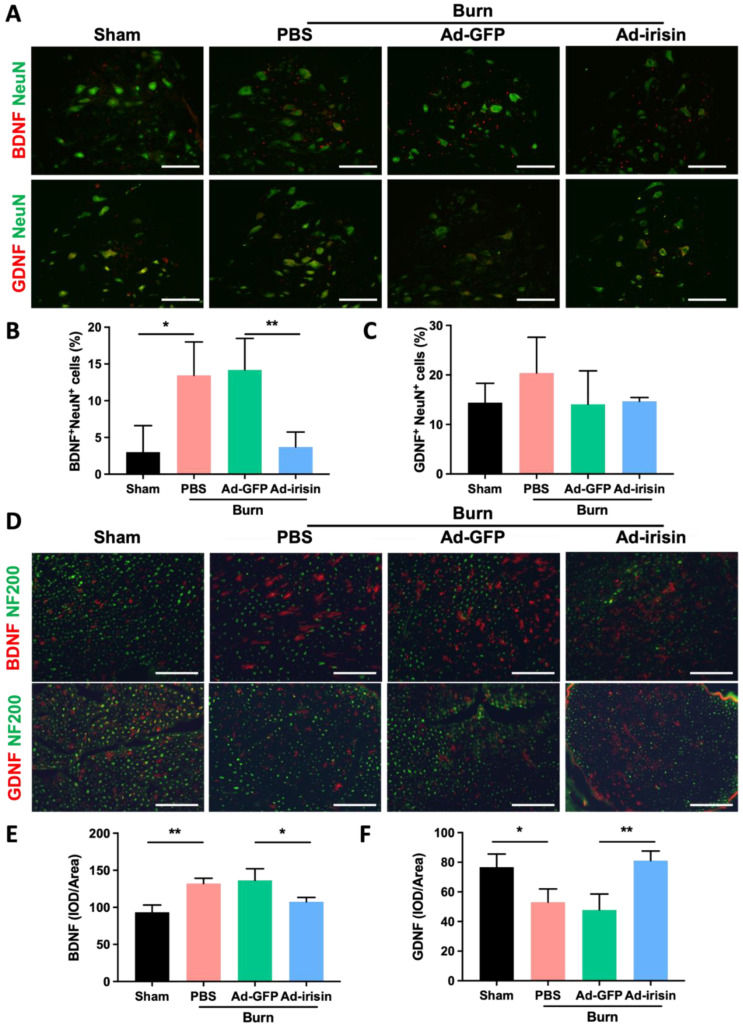
Spinal irisin gene delivery modulated BDNF and GDNF expression in the ventral horn and sciatic nerve after burn injury. (**A**) Immunofluorescence staining of BDNF and GDNF in the ventral horn in the fourth week after burn injury. NeuN staining was used to visualize neuron cells. (**B**,**C**) Representative bar graphs illustrating the ratios of BDNF+NeuN+ and GDNF+NeuN+ cells. Error bars represent standard deviations (SDs). * *p* < 0.05, ** *p*< 0.01, unpaired *t*-test. (**D**) Immunofluorescence staining of BDNF and GDNF in the sciatic nerve in the fourth week after burn injury. NF200 staining was used to visualize neurofilaments. (**E**,**F**) Representative bar graphs illustrating averaged optical intensity of BDNF and GDNF. Error bars represent SDs. * *p* < 0.05, ** *p* < 0.01, unpaired *t*-test.

**Figure 6 ijms-23-15899-f006:**
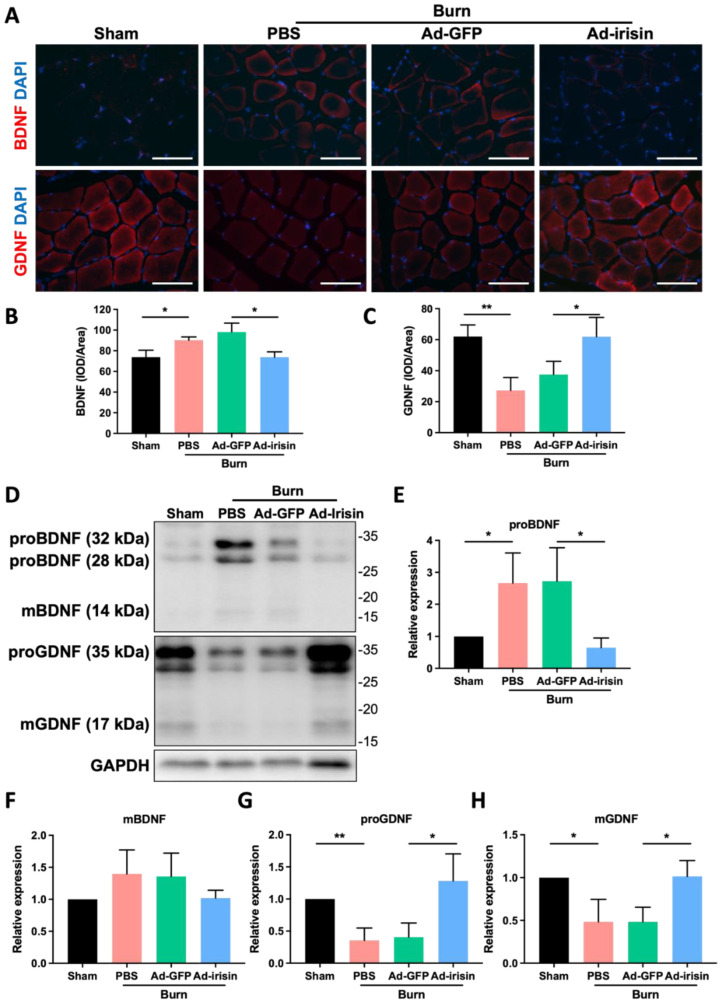
Spinal irisin gene delivery modulated BDNF and GDNF expression in ipsilateral gastrocnemius muscle after burn injury. (**A**) Immunofluorescence staining of BDNF and GDNF in ipsilateral gastrocnemius muscle in the fourth week after burn injury. DAPI was used to visualize the nucleus. (**B**,**C**) Representative bar graphs illustrating averaged optical intensity of BDNF and GDNF. Error bars represent standard deviations (SDs). * *p* < 0.05, ** *p* < 0.01, unpaired *t*-test. (**D**) Tissue of ipsilateral gastrocnemius muscle in the fourth week after burn injury was subjected to immunoblotting analysis. (**E**–**H**) Representative bar graph illustrating normalized expression of proBDNF, mBDNF, proGDNF, and mGDNF, with GAPDH as an internal control, in ipsilateral gastrocnemius muscle in the fourth week after burn injury. Error bars represent SDs. * *p* < 0.05, ** *p* < 0.01, unpaired *t*-test.

## Data Availability

Not applicable.

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
