# Peer review of "Spinal Irisin Gene Delivery Attenuates Burn Injury-Induced Muscle Atrophy by Promoting Axonal Myelination and Innervation of Neuromuscular Junctions"

_ijms, 2022, doi:10.3390/ijms232415899_

Round 1
Reviewer 1 Report
Further to their previous work, the authors have provided sound evidence showing the ameliorating effects of spinal cord delivered irisin in axonal myelination and neuromuscular junction innervation in burn-induced injury. The manuscript is well written and merits publication.
Here are some suggestions for the improvement of the draft.
1. BDNF has been well documented to be a major activator of mTOR. It has also been reported that BDNF promotes myelin formation via the neurotrophin receptor p75 (Tolwani 2004). However, in this manuscript, the authors observed the attenuation of BDNF and p75NTR, and increase in mTOR by irisin, along with its promotion on myelination and innervation. There is an obvious conflict with the results of other researchers. The author have noted the complication of that issue in the discussion section, however, due to the importance it has to the manuscript, further explanation on the issue will be helpful.
2. Figure 4(D) When observing innervation at the neuromuscular junction, we are expecting to see the overlapping of signals from neurofilament and acetylcholine receptor (α-BTX). Due to the resolution of the images, that couldn’t be done for all cells.
3. For the bar charts, the n numbers were not given for the statistical analysis, which is an important information about the rigor of the data. For Western blots, it is the number of independent replicate samples, while for cell images, it is the number of cells or the number of images being analyzed.
4. Materials and Methods, no sources of antibodies were given.
5.Figure 1 (c) 142 glycogen detection kit, no mention in the method section,
6.Line 121 what is ‘notexin’?
7. Line 34, could be a question of the English language, does the authors mean “… attenuate…sciatic nerve demyelination and reduction in NMJ innervation”?
Author Response
To Whom It May Concern
Please see the attachment.
Warm regards!
Sincerely yours,
Shu-Hung Huang, MD., PhD.
Professor
College of Medicine
Kaohsiung Medical University
Kaohsiung 807, Taiwan
Email: huangsh63@gmail.com
Tel.: +886-7-3121101 (ext. 5536)

Reviewer 2 Report
The authors perform very interesting experiments to demonstrate that spinal irisin reverses the neuromuscular effects of a burn. The studies are well-done, but I have several serious concerns about the model and applicability of the studies:
1) A lot of the introduction is very over-stated and exaggerated as to the significance of muscle atrophy and neurologic damage related to a burn. I have managed burns for 34 years and there is not a >50% incidence of prolonged neuromuscular problems in burns. A lot of the treatments described (growth hormone, IGF-1, etc.) are really related to the hypermetabolic changes and immobility of massive burns.
2) Another major problem is the relevance of the burn model to the above-stated complications of massive burn patients. The authors use a footpad burn, roughly 1-2% total body surface area, to recreate the clinical findings in patients with over 40% burns in people. While such small burns are common in people, they rarely cause muscle atrophy or neurologic problems. There are no systemic effects of such a small burn, so there is really no translation to the neuromuscular problems in massive human burns. One must wonder if the effects that they see in this model is from immobility or the rat's avoidance of bearing weight. I would suspect that a good control would be to splint an uninjured leg and they would have similar results. A lot of human research also focuses on the role of exercise in reducing long-term complications - would exercise change some of these early effects? Finally, the timing of the analysis (1 week after injection) is really not relevant to the changes seen months later in humans.
3) How do the authors propose to treat patients with intrathecal irisin?
4) A minor point - during the acute phase of burn injuries, there is no distributive shock, just hypovolemic shock. Distributive shock comes later.
Author Response
To Whom It May Concern
We appreciate you kind and precious comments to our manuscript. Following reply are our responses to your comments.
Please see the attachment.
Warm regards!
Sincerely yours,
Shu-Hung Huang, MD., PhD.
Professor
College of Medicine
Kaohsiung Medical University
Kaohsiung 807, Taiwan
Email: huangsh63@gmail.com
Tel.: +886-7-3121101 (ext. 5536)

Round 2
Reviewer 2 Report
Accept